# The Future of Sustainable Packaging: Exploring Biodegradable Solutions Through Extrusion, Thermo-Expansion, 3D Printing and Supercritical Fluid from Agro-Industry Waste

**DOI:** 10.3390/foods14234027

**Published:** 2025-11-24

**Authors:** Lacan S. Rabelo, Fabrício C. Tanaka, Sidney S. dos Santos, Fauze A. Aouada, Márcia R. de Moura

**Affiliations:** 1Grupo de Compósitos e Nanocompósitos Híbridos (GCNH), Department of Physics and Chemistry, Ilha Solteira School of Engineering, São Paulo State University (UNESP), Ilha Solteira 15385-000, SP, Brazil; lacan.rabelo@unesp.br (L.S.R.); sidney.souza@unesp.br (S.S.d.S.); fauze.aouada@unesp.br (F.A.A.); marcia.aouada@unesp.br (M.R.d.M.); 2Faculty of Animal Science and Food Engineering (FZEA), University of São Paulo (USP), Av. Duque de Caxias Norte, 225, Pirassununga 13635-900, SP, Brazil; 3Faculdades Integradas de Três Lagoas (FITL/AEMS), Av. Júlio Ferreira Xavier, 2750–Industrial District, Três Lagoas 79610-320, MS, Brazil

**Keywords:** biodegradable foams, thermoforming, extrusion, supercritical fluids, agro-industry waste

## Abstract

Due to environmental disasters caused by the use of plastic packaging, particularly expanded polystyrene (EPS), there is an urgent need to identify sustainable alternatives. Biodegradable foams derived from renewable polysaccharides have emerged as highly promising candidates to replace EPS, given their comparable cushioning and barrier properties. However, despite the rapid growth of research in this area, there has not yet been a comprehensive review addressing biodegradable foams as a specific class of packaging materials, particularly regarding their processing routes, raw materials, and functionalization. This work discusses conventional techniques for producing biodegradable foams, such as thermoforming and extrusion, as well as innovative methods, including supercritical fluids and 3D printing. It also examines key renewable polysaccharides and the incorporation of agro-industrial residues into foam matrices, aiming to improve performance and reduce costs. Furthermore, the article highlights advances in composite and nanocomposite foams, with particular emphasis on active properties such as ethylene absorption and antimicrobial activity capable of extending food shelf life. By directing attention to biodegradable foams as substitutes for expanded polystyrene, this review provides a unique contribution, filling a critical gap in the field and offering a foundation for future studies aimed at developing scalable, low-cost, and eco-friendly alternatives to plastics.

## 1. Introduction

Due to their versatility, low cost, and desirable barrier properties for food packaging, polymers derived from petroleum-based raw materials are manufactured and widely marketed. According to Walker and Fequet [1], plastic production has increased 20-fold over the last 50 years, reaching 368 million metric tons in 2019, with an expected two times increased growth rate in the next 20 years. Debnath et al. [2] report that plastic production increases by approximately 4% annually, with a significant portion, approximately 146 million metric tons, accounting for roughly half of global plastic production, coming from the packaging industry.

Considering the emission of toxic byproducts resulting from the recycling process and its natural degradation period, which can extend to about 400 to 500 years. This is particularly concerning given that an average person can use approximately 350 bags in a year, and the fact that the vast majority of countries worldwide lack adequate infrastructure for proper material management. This exacerbates recurring problems associated with plastics in nature, such as reduced soil fertility, threats to marine life, and contamination of groundwater [2,3,4].

In spite of the efforts made by experts and researchers to alter production processes or discover more efficient and sustainable approaches for a less harmful plastic packaging supply chain [5,6,7], the sheer volume of plastic waste in our oceans has reached alarming levels. It is now clear that responsibility cannot rest solely on citizens and researchers: companies and governments must also take action. Currently, 56 major global brands are responsible for more than 50% of plastic waste [8]. Without new policies and stronger corporate investment in sustainable packaging, reversing this scenario will be nearly impossible [9,10,11]. Additionally, recent studies have linked microplastics already present in water to intoxication and other health issues [1,12]. Considering our current global infrastructure, we can only recycle or reuse a mere 9% of the plastic produced [3,4]. The continued growth of conventional plastic packaging will further aggravate its environmental impact.

Although these general issues of petroleum-based packaging are already widely discussed in the literature, one specific category of expanded packaging, particularly expanded polystyrene (EPS, “Styrofoam”), remains underexplored. EPS is valued for its light weight, thermal insulation, and low cost [13], but it creates unique challenges: recycling is inefficient, its large volume burdens landfills [14], and it rapidly fragments into micro- and nanoparticles that disperse easily in ecosystems [15]. Despite being one of the most visible components of urban and marine plastic pollution, systematic reviews have rarely addressed EPS as a dedicated focus.

A more viable path forward involves the development of biodegradable foams capable of replacing EPS. Derived from renewable polysaccharides, these foams can provide comparable cushioning and barrier properties while offering biodegradability [16,17]. Despite its small size and its representation as a minute fraction of the market, the biodegradable and compostable packaging industry is introducing new options and products based on biodegradable foams with barrier properties that mimic expanded isoprene [18,19].

Biodegradable foams can be produced using various techniques, including extrusion, thermoforming, microwave processing, injection, lyophilization/solvent exchange, and supercritical fluid extrusion. However, due to factors such as capacity, speed, and production cost, extruders and thermoforming (baking) remain the most common methods adopted by companies and researchers [19,20]. The formation of foam structures in processes involving heating is based on the development of a porous structure, which occurs during the evaporation of solvents and/or water vapor from a polymer mass with high moisture content. Under these conditions, the kinetic energy of the generated vapor serves as an expansion agent. The production of foams used in packaging through thermo-expansion is exemplified in Figure 1, along with an illustrative video accessible via the QR code in the bottom left corner of Figure 1 [20].

In the literature, two main works provide surveys of methods for obtaining biodegradable forms. The work developed by Tapia-Blácido and collaborators [21] focuses on starch as the primary raw material for these packages. It also discusses the influence of parameters such as gas dissolution, cell nucleation, temperature, pressure, and other physicochemical properties on the final characteristics of the packaging. This work also explores the combination of starch with other materials, the production of bioactive packaging, and its applications and challenges in the market. Other topics discussed in this review include strategies and improvements in mechanical properties, processing, moisture resistance, and antimicrobial activity of packaging made from starch foams.

On the other hand, the work written by Mali [19] also emphasizes the use of starch as one of the most commonly employed raw materials for this type of packaging. However, this work delves into a more comprehensive analysis of production methods for packaging, particularly emphasizing extrusion and baking using thermoforming. Additionally, it provides a detailed approach to foams obtained from polymeric composites, blends, and other foams formed by copolymers in conjunction with other biodegradable polymers, notably Polyvinyl alcohol (PVA) and Polylactic acid (PLA). The paper also explained the influence of formulation and techniques on water sorption capacity as well as morphological, mechanical, and barrier properties.

This work aims not only to discuss the methods and processes that influence foam formation in biodegradable packaging, but also to provide an up-to-date survey of polysaccharide-based raw materials, with emphasis on agro-industrial residues capable of improving barrier performance and/or reducing production costs. Within this scope, particular attention is given to emerging strategies such as 3D-printing-assisted foaming, the incorporation of waste-derived fillers, and the development of composite and nanocomposite foams to produce more competitive materials.

In this context, we treat biodegradable polysaccharide-based foams as a distinct class of materials explicitly designed to replace expanded polystyrene (EPS) in food packaging. Unlike earlier reviews, which typically focus either on starch foams or on bio-based packaging in general, the functional requirements and environmental drawbacks of EPS are taken here as the foundation of a unified framework for comparing the main foaming routes (thermoforming, extrusion, 3D printing, and supercritical fluid technology). We also synthesize recent advances in foams reinforced with agro-industrial residues and other polysaccharides and link these technological developments to a critical discussion of economic, infrastructural, and regulatory barriers to large-scale adoption.

## 2. Methods Used for the Production of Biodegradable Foams

There are several methods employed for the production of biodegradable foams based on polysaccharides. Among the most consolidated approaches are thermoforming and extrusion, which remain the backbone of large-scale industrial processes [22,23,24]. More recently, innovative technologies such as microwave-assisted processing [20,25], supercritical fluid application [26,27], and assistive manufacturing (3D printing) [28,29] have emerged, aiming to reduce production costs, shorten processing time, or enable the design of foams with higher structural complexity compared to conventional routes.

Fabrication parameters such as pressure, temperature, and the method of acquisition play a decisive role in determining the final performance of foams, directly influencing their density, expansion ratio, water vapor permeability, mechanical strength, and thermal stability. among others [30,31,32]. Beyond the intrinsic physicochemical properties of the selected polymeric matrix, several extrinsic factors must also guide the choice of processing route, such as equipment cost, manpower qualification, process scalability and production capacity [33].

To highlight the trade-offs involved in selecting each technology, a comparative matrix was developed (Table 1). This table summarizes the advantages and limitations of the four manufacturing methods most frequently reported in the literature. Based on references [19,22,24,34,35,36,37,38,39], the authors considered objective indicators reported in the table and in other works cited in this article. Accordingly, the classifications were derived from a composite index divided into the ranges: Low, Medium, Good, and High. Extrusion has been employed for this type of application since 1980, and both extrusion and thermoforming are already used in large-scale production of such materials, whereas 3D printing and supercritical fluid methods are not. It should also be noted that extrusion requires the use of plasticizers during processing, while thermoforming does not. In terms of processing time, 3D printing is the most time-consuming technique, requiring hours to produce a single part compared with the minutes needed for thermoforming and extrusion. Finally, it is important to highlight that no industrial-scale equipment is yet available for supercritical fluid processing, only laboratory-scale devices. According to Zhou, Tian, and Peng [35], however, the acquisition of such equipment would involve a considerably higher installation cost than other methods. In the context of the Limitations criteria, the different classifications refer to the severity of the constraints inherent to each manufacturing method. A Low limitation score indicates that the technique presents minimal operational or material constraints and can be implemented with few technical barriers. A Medium classification denotes the presence of relevant restrictions, such as moderate sensitivity to processing conditions or partial incompatibility with certain biopolymer systems—which may require optimization but do not substantially hinder applicability.

Before selecting a fabrication technique, researchers must carefully consider factors such as the nature of the polymer and additives, the gas retention capacity within the foam structure, and the rheological behavior (viscosity) of the formulation. Consequently, the literature reports a broad range of methodologies, parameters, and compositions tailored to different applications.

This review does not aim to provide an exhaustive description of methods but instead focuses on the most widely used approaches and experimental parameters that have been recurrently applied in the context of food packaging. To contextualize the evolution of these technologies, Figure 2 presents a timeline of relevant milestones in the development of biodegradable foams and their integration into food packaging [19,36,40,41].

### 2.1. Thermoforming

Thermoforming, also known as baking, is a widely adopted method among companies and researchers. This technique involves using heat and pressure to shape a specific polymeric matrix into a predefined form or mold. Some authors liken thermoforming to the process of making waffles. During the thermoforming process, the polymeric mass placed within the mold is exposed to temperature and pressure conditions exceeding those of water evaporation. Under these circumstances, water vapor possesses high kinetic energy, leading to collisions with the foam’s polymeric network, resulting in a porous and solid structure after the water vapor evaporates [19,42,43,44]. Under these circumstances, water vapor possesses high kinetic energy, leading to collisions with the foam’s polymeric network, resulting in a porous and solid structure after the water vapor evaporates [19,42,43,44]. The diagram and video embedded in the QR code of Figure 1 illustrate the processes occurring in thermoforming.

After the 1990s, thermoforming offered the market a new option for obtaining packaging with more complex shapes and structures, which would be very difficult to manufacture using the traditional extrusion method. It also provided excellent cost-effectiveness, capable of producing large quantities of packaging. However, according to Chakraborty and colleagues, thermoforming comes with some disadvantages, such as the need for specific equipment and molds that can increase the cost of the process, as well as the requirement for skilled labor and lower production speed when compared to other production methods [19,43].

Despite the disadvantage of thermoforming requiring molds that can increase equipment costs, this technique can offer a greater variety of shapes with higher refinement compared to extrusion. The processes and minimum ingredients necessary for this technique are the same as those used by Tapia-Blacio and colleagues [21] to explain the starch foam production system. These include gas dissolution in a polymer solution, cell nucleation due to system saturation caused by increased temperature or reduced pressure, cell growth, and foam stabilization with the solidification of the polymer structure. Although this technique is more flexible, factors such as molecular weight, viscosity, component dispersion, and crystallinity still influence the final product characteristics. However, the amount of material used will depend on the polymer matrix and its rheological properties. Foams manufactured by thermoforming tend to have a denser, closed external layer with lower porosity compared to the interior of the material, which has larger pore diameters [19,21].

In studies conducted by Debiagi Marjorie and Mali [45], new biodegradable packaging made of foams composed of PVA and cassava bagasse was developed. This study revealed that the investigated concentrations with lower susceptibility to water were those with the lowest amounts of PVA in their structure. However, it was also observed that the addition of PVA to the structure led to an increase in the tensile strength values of the samples. It went from 19.49 MPa with 0% PVA to 206.89 MPa with 7.5% PVA. In another study conducted by Vercelheze and co-authors [46], foams based on cassava starch, sugarcane bagasse fiber, and sodium montmorillonite clay were manufactured through thermoforming. Despite the successful incorporation of these components and the formation of the packaging, the samples exhibited a water absorption capacity exceeding 50%, rendering them unsuitable for beverages and high-moisture foods. However, according to the authors of the study, the material could be utilized as an alternative packaging for low-moisture content foods.

### 2.2. Extrusion

During the extrusion process, in order to create polymeric materials, the extrusion conditions must heat the material sufficiently to induce its melting. As such, extrusion is a technique that relies on high-temperature conditions, pressures, and shear rates. It has been widely used in the manufacturing of various materials, including new biodegradable packaging based on biopolymers. To obtain biodegradable packaging through extrusion, the use of plasticizers is very common. However, their addition can affect the melting temperature range of polymeric matrices, necessitating further studies on the parameters used during processing [19].

The production of biodegradable foams through extrusion can be performed using single-screw and twin-screw extruders. The cheaper and more commonly used option is the single-screw extruder, while twin-screw extruders are more expensive and used for higher-yield productions. A High Temperature Short Time (HTST) extrusion process for polysaccharide-based foams can be conducted at temperatures ranging from 120 to 180 °C and screw rotation speeds between 70 and 400 rpm, depending on the material. The extrusion process can be divided into feeding, mass transport, flow through the mold, and post-mold treatment. During the process, the polymer mass is mixed with water to achieve a moisture content between 13 and 30%, plasticizers (usually sorbitol or glycerol), and other additives. The mixture is then heated to its melting point, and through rapid decompression, bubble formation and foam nucleation occur [19,21,47].

Vorawongsagul, Pratumpong, and Pechyen [48], synthesized new composite foams made from cellulose fibers extracted from kraft paper, supported by networks of PLA/poly(butylene succinate) (PBS). The extrusion process occurred within temperature ranges of 180–200 °C. Despite the increased affinity with water molecules caused by the presence of water, an increase in the concentration of cellulose fibers can be observed, leading to an enhancement in the thermal and mechanical properties of the foams with fiber insertion. In other studies, such as Brant et al. [49] and Lopez-Gil and colleagues [42], a combination of extrusion and radiation (ionizing radiation and microwave radiation, respectively) was used to create foams consisting of starch or starch with natural fibers. In the first case, the use of ionizing radiation can contribute to the formation of new cross-linked structures with increased mechanical strength and barrier properties. On the other hand, the use of microwaves in conjunction with extrusion provides uniform heating of water molecules, leading to thermal expansion, making it one of the most promising techniques for obtaining biofoams. Thus, despite being a technique used for decades, new updates and methods used in conjunction with extrusion can be very useful for researchers and companies aiming to study and develop new biodegradable packaging made from polysaccharide foams.

### 2.3. Three-Dimensional Printing

Layer-by-layer deposition using 3D printing enables the creation of geometries and structures with a higher level of complexity, facilitating the development of more interactive and customizable designs. Similarly to thermoforming, this technique also requires highly skilled labor. Another disadvantage of 3D printing is the slow printing speed for the production of complex objects, which makes it impractical for large-scale manufacturing [43]. Nida Moses and Anandharamakrishnan [44] were able to use banana peels and sugarcane bagasse (two non-printable materials), along with a guar gum base (a printable material), and 3D printing to develop new food packaging based on polymeric blends with properties and characteristics capable of competing with expanded polystyrene. The study demonstrated that it is possible to blend low-cost materials that can be reused to create new packaging by forming stable blends with raw materials used for 3D printing of biopolymers. According to Nida and colleagues [44,50], this study showcased a way to repurpose agro-industrial waste through 3D printing, a technique that could contribute to the development of new packaging with different types of properties, shapes, and structures that may replace products made from synthetic polymers.

The most commonly used 3D printing technique for biopolymers is extrusion-based printing. Therefore, the parameters and concepts used are very similar to those discussed in the previous section. However, additional parameters such as nozzle temperature, bed temperature, extrusion speed, and layer height must also be considered to achieve a higher-quality print [50,51,52].

### 2.4. Supercritical Fluid

Another way to obtain foams with porous structures is through the application of supercritical fluids at high pressures. When the system is depressurized, this processing method requires less energy and allows the use of heat-sensitive raw materials without thermal degradation occurring [39]. Up to now, supercritical carbon dioxide is the most widely used fluid for this type of application. Most studies that use supercritical fluids choose carbon dioxide because of its low cost, easy availability, and favorable processing conditions [53,54].

Regarding the proposal to obtain new biodegradable foams, synthesized through renewable and cost-effective sources capable of competing in the market with polystyrene, where supercritical carbon dioxide fluid was used as a fabrication method, it is worth highlighting the work developed by Pereira da Silva et al. [55]. In this study, the influence of adding pupunha fiber obtained from agro-industry waste on the mechanical, morphological, and rheological behavior of foams supported by poly(butylene adipate-co-terephthalate) (PBAT) polymeric matrices was investigated. The composites were synthesized by mixing PBAT with the respective pupunha fiber concentrations using a micro-extruder. The mixture took place under a nitrogen atmosphere, and then the samples were injected into their respective molds. Subsequently, to form the foams, the material was subjected to a high-pressure reactor with a capacity of 300 mL, subjected to a pressure of 100 bar of CO_2_. In this study, it was found that samples with 30% fiber exhibited more promising mechanical properties for use as food packaging, demonstrating the possibility of obtaining cheaper and more environmentally friendly and economical biodegradable materials.

## 3. Raw Materials

In general, biodegradable polymers are formed by polysaccharides, proteins, and other macromolecules. In the presence of fungi and bacteria and when introduced into bioactive environments conducive to enzymatic catalysis or chemical hydrolysis processes, these polymers have their macrostructures reduced to carbon dioxide, water, biomass, methane, and other substances. Other desirable characteristics for the use of these materials such as raw materials for packaging include low toxicity, abundance in nature, and consequently low cost, biocompatibility, and ease of processing. This section does not aim to discuss all materials used for the development of biodegradable foams; however, it will address some of the main raw materials employed, studied, and obtained from cheap and renewable sources for the production of packaging composed of biodegradable foams.

### 3.1. Cassava Starch with Residue Fibers

Cassava is already a widely used raw material for producing biodegradable foams, with cassava starch being one of the most utilized polysaccharides in the production and development of packaging made from biodegradable and compostable foams. Nevertheless, by 2025 only seven studies using the keywords “starch, food, packaging, biodegradable, and foams” had been published. It can be seen in Figure 3 that the first article appeared in 1996, and since 2020 the field has produced an average of approximately four articles per year, indicating that it remains a relatively underexplored research area.

However, controversy arises among some researchers and experts who question the use of a component for which high demand could create competition with starch intended as an additive in the food industry, potentially causing supply problems or affecting the value of this component in the sector [56,57]. For example, in the case of a high demand for the production of biodegradable packaging, since starch has promising properties for this type of application, meeting large-scale production would require a significant amount of starch extracted from food sources such as corn, cassava, and potatoes. This could lead to competition with the food market for this raw material. On the other hand, starch is one of the most abundant and renewable natural polymers on the planet. Its low cost and desirable characteristics make it suitable for large-scale production [47].

For this study, a keyword co-occurrence analysis was conducted using VOSviewer 1.6.20 software. An advanced search on ScienceDirect was performed using the terms ‘biodegradable foams and food packaging,’ filtering for works conducted from 2022 to the present. From the 1636 works initially presented, only research articles were selected, and 100 articles were initially separated. Figure 4 illustrates the results of the co-occurrence analysis, indicating that among these 100 articles, the terms ‘active packaging’, ‘chitosan’, ‘cellulose’, ‘starch’, ‘biodegradable’, and ‘biodegradability’ carried the most weight.

The fact that the term “starch” is associated with “cellulose” may stem from studies concentrating on incorporating cellulose fibers derived from agro-industrial waste into the structure of biodegradable foams. The objectives include reducing the production cost of these materials and enhancing various properties such as mechanical strength, thermal stability, and resistance to water [58]. Another reason for the presence of these two polysaccharides in this analysis could be linked to the fact that these components and their derivatives are among the most commonly used raw materials for the production of foams and other biodegradable packaging. Hence, the authors of this study opted to emphasize research focused on utilizing these components in conjunction with the integration of agro-industrial waste into the composition of biodegradable foam-based packaging.

Starch is a polysaccharide primarily composed of amylose chains, consisting of linear structures of D-glucose connected by α 1,4 glycosidic bonds, along with amylopectin, another macromolecule composed of D-glucose with chains connected by α 1,4 glycosidic bonds and branched by D-glucose through α 1,6 bonds. Starch serves as an energy reserve in plants and is commonly extracted from sources such as corn, rice, cassava, and others, in the industry [57].

Among the most promising articles involving foams made from starch and fibers extracted from plant residues, studies that utilized cassava starch achieved particularly favorable results. The literature review conducted for this work uncovered a study that incorporated commercial cellulose fibers [58], fibers extracted from coconut waste [59,60], peanut skin [61] and sesame cake (a byproduct of the oil industry) [62].

Although these studies demonstrate progress, comparisons among different agro-industrial residues remain fragmented. A critical analysis ranking these fibers in terms of cost-effectiveness, reinforcement capacity, and water-resistance improvement would offer clearer guidance for industrial adoption [63,64,65].

Among the recent works presented in the literature on the incorporation of fibers into the structure of starch-based foams, it is worth mentioning the study conducted by Engel, Luchese, and Tessaro [57]. The primary objective of this research was to assess the addition of pre-gelatinized starch (a byproduct formed and discarded in starch production).

In the analysis of mechanical properties, it was observed that the presence of pre-gelatinized starch resulted in an increase in tensile strength, along with a rise in elongation values. The addition of fiber from the inner peel also yielded promising results, such as an 84% reduction in the packaging’s water absorption capacity, demonstrating the potential to decrease production costs while obtaining materials with more favorable properties.

Despite these promising laboratory-scale results, challenges such as fiber dispersion, moisture control, and reproducibility across batches remain insufficiently addressed in the literature. These factors constitute major bottlenecks to scaling up cassava starch foams reinforced with residues for industrial production [64,65,66]. These factors constitute major bottlenecks to scaling up cassava starch foams reinforced with residues for industrial production [65].

There are already packaging materials composed of cassava starch on the market; however, these materials still have certain disadvantages related to price, mechanical properties, and water susceptibility. The addition of cellulose fibers extracted from plant residues can improve some properties, such as tensile strength and others related to the material’s interaction with water.

Therefore, incorporating agro-industrial fibers into cassava starch foams presents a dual opportunity: valorizing residues while mitigating the intrinsic limitations of starch. However, the absence of standardized testing protocols and inconsistent reporting of mechanical and barrier properties (as evidenced in Table 2) limit cross-study comparability and hinder practical recommendations for industry [64,67]. Table 2 illustrates examples of studies that incorporated plant fibers into cassava starch foams. The table also provides values for mechanical and hydrophilic analyses of these foams.

### 3.2. Cellulose Derivatives-Based Foams

Cellulose is the most abundant polymer on the planet and can be extracted from various types of plant-based raw materials. It is composed of linear chains of D-glucose linked together through β 1,4 glycosidic bonds [68]. As presented in Figure 4, along with starch, cellulose is one of the most explored polymers for the production of biodegradable foams. The use of this polysaccharide as the foundation for new biodegradable foams can provide promising alternatives to replace polystyrene. Results obtained by Ottenhall A, Seppänen T, and Monica E. K [69] show that it was possible to obtain a material composed of cellulose fiber foams, cationic chitosan, and polyvinyl amine with good water stability and inhibition properties against *Aspergillus brasiliensis* and *Escherichia coli*. These results indicate a promising use of cellulose not only for replacing petroleum-derived plastic packaging but also for demonstrating the possibility of producing active and intelligent foams with low production costs.

Another method of utilizing cellulose for the production of new foams involves applying cellulose derivatives, which can streamline the manufacturing process for these innovative materials. The study conducted by Miranda-Valdez and colleagues [70] also resulted in foams exhibiting antimicrobial properties, showing inhibitory effects against *Escherichia coli*. In this study, foams were derived from methylcellulose, cellulose fiber, and lignin.

Another investigation exploring the use of cellulose derivatives is the study conducted by Karlsson et al. [71], where new foams composed of hemicellulose and hydroxypropyl methylcellulose were successfully obtained. In a separate work by Karlsson and collaborators [72], it was feasible to produce foams using hydroxypropyl methylcellulose and ethyl hydroxyethyl cellulose, demonstrating satisfactory density and characteristics through hot milling.

Thus, the discussion in this section highlights that cellulose derivatives and cellulose fibers extracted from residues can serve as a cost-effective and renewable option as raw materials or reinforcing agents for new studies aimed at developing foams for applications such as food packaging.

Despite the promising antimicrobial and structural improvements, the variability in cellulose sources and chemical modifications makes direct comparison across studies challenging. Few works report standardized mechanical, barrier, or degradation tests, which limits scalability assessment. Future studies should prioritize harmonized testing protocols and life-cycle analyses to validate the industrial potential of cellulose-based foams [64,70].

### 3.3. Other Polysaccharides Used in the Synthesis of Foamed Packaging

Natural polymers, such as pectin [73] and chitosan [74], are also used as the basis for biodegradable foams. However, due to the cost of these raw materials and the fact that they are less abundant than starch and cellulose, they are less employed compared to these two polysaccharides.

In studies conducted by Ottenhall, Seppänen, and Monica [69] and Machado et al. [62], these components appear in the minority compared to starch and cellulose; however, their presence can enhance properties such as tensile strength, among others. Nevertheless, despite not being as widespread, these polysaccharides are also renewable and abundant in nature, offering the potential for new materials with promising properties in applications such as food packaging.

As seen in the case of packages composed of PLA and chitosan, obtained through the freeze-drying method described by Mania and colleagues [74], it was reported that the addition of chitosan conferred bactericidal and bacteriostatic properties to the material against *Escherichia coli* and *Staphylococcus aureus*.

These studies highlight the potential of minor polysaccharides, the high production cost and limited supply chain for pectin and chitosan remain significant barriers. It is also prudent to emphasize techno-economic evaluations and exploring waste-derived sources of pectin and chitosan (citrus peel residues, crustacean shells) could increase their relevance in the packaging sector.

### 3.4. Protein Used in Biodegradable Foams

Regarding studies that use proteins as raw materials for the production of foam-based packaging from biopolymers, only two studies have been reported. The first, by Salgado et al. [75], describes the synthesis of a biofoam composed of cassava starch, sunflower protein, and cellulose fibers. In this study, the addition of cellulose fibers was primarily aimed at enhancing mechanical properties, barrier properties, and water resistance.

The second study, by Jarpa-Parra and collaborators [76], produced foams from lentil protein and cellulose fibrils, which exhibited an irregular structure and non-uniform porosity.

The authors of this study believe that the low number of studies focused on using proteins for developing new biofoams is due to proteins being macronutrients. Those with significant nutritional value, owing to their diversity of essential amino acids, also have high market value and demand, especially in the fitness product market. Therefore, using this type of biopolymer could increase the production cost of the packaging, raising its price and making it less competitive with other packaging options in terms of market value. While the limited number of studies reflects concerns about cost and food competition, proteins may still hold strategic potential if sourced from industrial byproducts (Whey, Corn Zein) [77,78].

## 4. Composites and Nanocomposite Foams

For the discussion focused on the use and development of new foams for food packaging, another keyword co-occurrence analysis was conducted using VOSviewer with the terms ‘starch foams nanocomposites, and food packaging.’ Articles from 2022 to the present were filtered, resulting in 139 outcomes, with only 100 articles being selected for correlation analysis. However, this search indicates that it is still a relatively unexplored option that can significantly contribute to this research field.

Figure 5 displays networks correlating the keywords of these articles, revealing three different networks, indicating weak correlations among the terms of selected works. Nevertheless, it is noteworthy that the terms ‘active packaging’ and ‘antimicrobial’ form separate networks; these results indicate that part of the research on nanocomposite foams has focused solely on incorporating new structures to make the resulting materials more promising for food-packaging applications. Another research direction aims to incorporate nanostructures not only to improve physicochemical properties but also to add value by enabling the development of new active and intelligent packaging capable of extending food shelf life. Such functionalities may rely on mechanisms such as atmosphere control (e.g., ethylene removal), modulation of gas permeation, or antimicrobial activity. In particular, nanostructured reinforcements such as nanoclays, nanocellulose, and carbon-based fillers not only enhance mechanical and barrier properties but also introduce functionalities such as antimicrobial action, antioxidant release, and gas scavenging capacity [79,80,81]. It is also notable that the spheres representing the correlation of articles with the keywords ‘active packaging’ and ‘antimicrobial’ appear in green and yellow, respectively, suggesting that studies employing these terms in the context of nanocomposite foam development are more recent than those centered on the use of biomass or agro-industrial residues in packaging formulations.

An example worth mentioning regarding the potential of foams obtained from structures formed by composite and nanocomposite materials is described by Nooun et al. [82]. In this study, new foams composed of natural rubber, rice starch, and activated carbon were synthesized. An increase in thermal and mechanical properties was observed with the addition of activated carbon to the material structure. Additionally, the foam gained the ability to absorb ethylene maturation gas, delaying ripening and extending the shelf life of ‘Hom Thong’ bananas, preserving pulp firmness for up to 6 days in ambient storage. Table 3 presents results and examples of studies related to the development of foams made from composites and nanocomposites. Despite the limited number of works found, this is still an area that has the potential to offer new products and solutions.

However, similar to biodegradable films and coatings, new syntheses of composite or nanocomposite foams could result in the development of new active and intelligent packaging. This study has shown that some materials already possess antimicrobial properties [69,82], and further research on active and intelligent biodegradable foams is forthcoming. Wang et al. [86] discuss methodologies used for producing active and intelligent packaging from polysaccharides. The authors of this study believe that it is feasible to adapt some of these methods to develop new biodegradable foams for packaging formulations. These foams could be used to monitor and detect changes in chemical composition, pH, moisture, oxygen levels, and enzymatic or microbial activity.

For example, incorporating pH-sensitive indicators, oxygen scavengers, or moisture sensors into starch–nanocellulose foams could transform them into affordable smart packaging. However, to progress beyond laboratory demonstrations, studies must also address storage stability, consumer safety, and industrial scalability [87,88]. This suggests the potential to create new types of packaging that are environmentally friendly, cost-effective, and provide added value and properties not offered by conventional packaging.

Finally, an analysis of the keywords “nanocomposites, food, packaging, and foams” was conducted in the Elsevier Scopus database, which revealed only twenty-two articles published up to 2025. As shown in Figure 6, after 2011, three studies related to this topic were published. On average, 1.1 studies have been published since 2022. This trend indicates that the development of biodegradable nanostructured foams remains underexplored. Moreover, most of the methodologies used to characterize these new materials vary considerably among authors, which makes it difficult to compare foams obtained from different raw materials or preparation methods. Consequently, this lack of standardization complicates the evaluation of their key properties. Nevertheless, the development of new composites and nanocomposites shows promising potential for creating innovative packaging with more competitive properties and costs.

## 5. Future Prospects and Challenges

As mentioned earlier in this study, one of the greatest challenges for packaging made from biodegradable foams is competing with conventional packaging in terms of price and barrier properties. However, some biodegradable packaging solutions are already breaking this barrier and finding their place in the market, such as those commercialized by Eco-foam^®^ (National Starch & Chemical, Bridgewater, NJ, USA), Clean Green (Starch Tech Inc., Golden Valley, MN, USA), Bio-solo (Indaco Manufacturing Ltd., Pickering, ON, Canada), Cornpol1 (Japan Corn Starch, Minato City, Toranomon, Japan), Bioflex1 (FKuR Kunststoff GmbH, Willich, Nordrhein-Westfalen, Germany), Mater-bi1 (Novamont, Novara, Piemonte, Italy), Biopac (Biopac Ltd., Worcester, Worcestershire, UK), Bioplast1 (Biotec GmbH, Germany Willich, Nordrhein-Westfalen, Germany), Ecoplast (Groen Granlaat, Veendam, Groningen, Netherlands), Vegemat1 (Vegeplast S.A.S, Bazet, Occitânia, France), and Solanyl1 (Rodenburg Biopolymers, Oosterhout, Noord-Brabant, The Netherlands) [36].

On the other hand, most companies that commercialize biodegradable packaging, including those made from foams, are located in and sell their products in developed countries. Due to the scale of this problem, such products should be marketed in more countries to generate new jobs. However, it is important to note that the cost of these products must remain low to encourage more people to switch from conventional to new packaging.

Another challenge lies in supply chain logistics and technology transfer. Developing countries often lack composting and industrial biodegradation infrastructure, which directly limits the adoption of bioplastics [89,90]. Policies promoting local production units, the valorization of regional raw materials, and subsidies for composting facilities should be highlighted as essential strategies to expand the global reach of biodegradable foams [91,92].

According to Ghasemlou, Barrow e Adhikari [93], one of the main challenges for this type of packaging in the market, in addition to the factors already presented in this study, is that many consumers and food companies are not convinced of the utility of bioplastics due to inconsistencies in their labeling and/or lifespan. Despite the significant growth in the bioplastics market, there is a lack of global facilities and infrastructure for their production and/or composting. Lastly, for greater safety and to validate the commercialization of bioplastics in food packaging, long-term biological effects and toxicity assessments must be meticulously studied. This requires qualified laboratories and researchers, making it challenging to discover new packaging solutions in less developed countries.

This challenge is compounded by disparities in regulatory frameworks across regions. For instance, while the European Union EN 13432 [94] and the United States ASTM D6400 [95] have established clear compostability standards, many countries in Latin America and Africa still lack equivalent certification pathways. Such regulatory gaps hinder international trade, foster consumer distrust, and delay the harmonization of global markets. Establishing unified standards and transparent labeling is therefore a priority [96,97].

Therefore, new public policies and incentives from large companies should be promoted to popularize and encourage the use of these new packaging solutions. Policies such as banning the use of certain plastic items, as has been performed in the Brazilian states of São Paulo and Rio de Janeiro (Law Project No. 3794/2018 and Law Project No. 631/2018, respectively), should be implemented. Other incentives, such as tax support and investments, could significantly contribute to increasing the market share of biodegradable packaging, which currently represents less than 1% of the market compared to conventional plastic packaging (370 million tons).

Beyond regulation, linking policies to measurable environmental benefits would further strengthen their impact. For example, replacing even 5–10% of EPS packaging with starch or cellulose-based foams could prevent thousands of tons of non-biodegradable waste from entering landfills and marine ecosystems each year. Moreover, case studies such as the bans on EPS in the European Union, California, and India demonstrate that regulatory pressure, when combined with financial incentives, can accelerate innovation and market adoption [91,92,98].

Increasing the supply and consumption of these new packaging solutions could help minimize the environmental damage caused by current packaging [36,93,99].

## 6. Conclusions

Currently, the availability of biodegradable packaging in the market is quite limited compared to conventional plastics. However, the emergence of new products and research focused on the development of materials, such as biodegradable and compostable foams, with the aim of replacing polystyrene (one of the most widely used plastics on the planet and, consequently, one of the most environmentally damaging) can contribute to making more environmentally friendly products available in the market and also help make such products increasingly accessible.

The use of raw materials derived from renewable and abundant polysaccharides, such as cellulose and starch, can contribute to the production of low-cost products. However, the incorporation of residues, such as fibers obtained from agro-industry, can favor the development of new packaging with low production costs and properties capable of competing in the market with conventional plastic packaging. The survey in this study also showed that, due to the versatility of these materials or even the addition of other nanostructured materials, foams with added value, such as microbial inhibition properties and even packages that slow down the aging of fruits, can be obtained, enabling the production of packages with greater market appeal.

Regarding manufacturing techniques, new technologies such as 3D printing and the use of supercritical fluids can enable the development of new research in a greater number of institutions and laboratories, and the production of packaging with lower energy consumption, respectively. However, the choice of technique, whether conventional ones like extrusion and thermoforming, or even new manufacturing methods, will depend on factors such as investment capacity, workforce qualifications, material type, and market demand, among others.

Thus, regarding the techniques discussed in this work and the new synthesis methods, such as the incorporation of fibers, the development of nanocomposites, and the design of active and intelligent packaging, future research should prioritize strategic investments and the development of novel production techniques. In particular, selecting appropriate raw materials and optimizing synthesis routes can support the creation of new biodegradable materials capable of minimizing environmental impact while competing economically with conventional plastics. Furthermore, these efforts may not only lead to low-cost and eco-friendly products but also enable the development of intelligent packaging with added value, such as systems capable of extending food shelf life and reducing microbial contamination. Such practical research directions can guide the advancement of more efficient production strategies, novel formulations, and combinations of manufacturing techniques that yield innovative and market-competitive packaging solutions.

In conclusion, based on the discussions presented in this article, it can be concluded that with an increase in investment in new research, it will be possible to increase the supply of bioplastics in the future. With this, it will be possible to provide packaging in larger quantities and at more affordable prices for a larger portion of the population, thereby reducing the environmental impact caused by packaging derived from petroleum derivatives.

## Figures and Tables

**Figure 1 foods-14-04027-f001:**
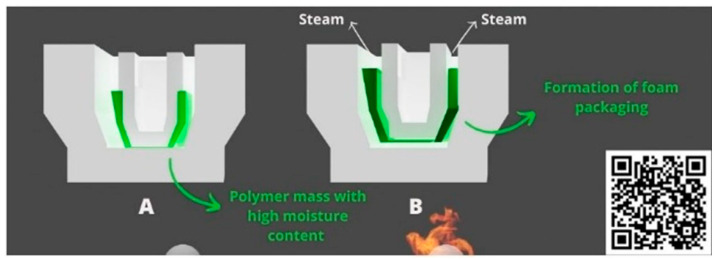
Representation of the process for obtaining foam packaging through thermo-expansion, in which (**A**) the pre-heated polymer mass is shown before expansion, and (**B**) the foam-structured polymer is formed as a result of thermo-expansion. Link to the video in the QR code: https://www.youtube.com/watch?v=JCMScX5vjSQ (accessed on 20 November 2025).

**Figure 2 foods-14-04027-f002:**
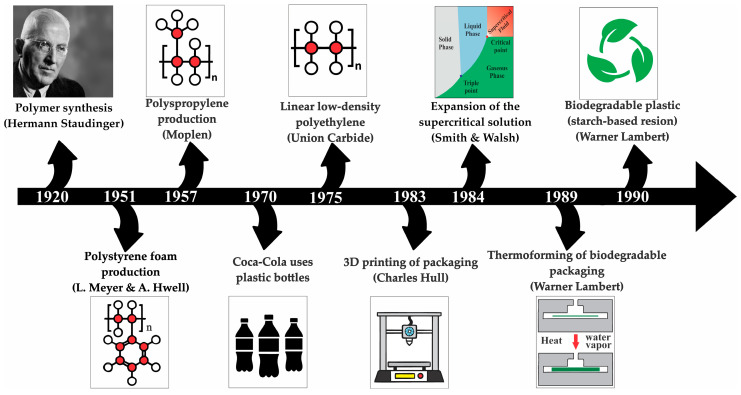
Timeline of significant technological milestones in the field of biodegradable foams and food packaging.

**Figure 3 foods-14-04027-f003:**
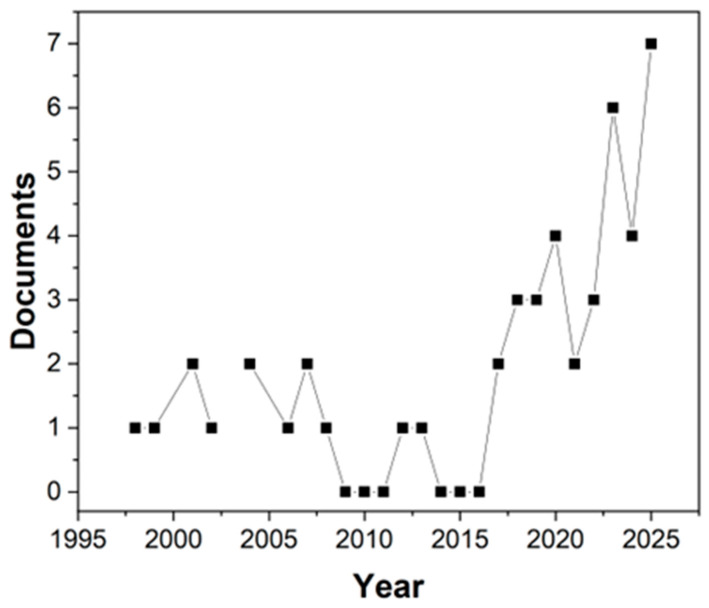
Graph showing the number of articles retrieved from Scopus using the keywords “starch, food, packaging, biodegradable, and foams” over time.

**Figure 4 foods-14-04027-f004:**
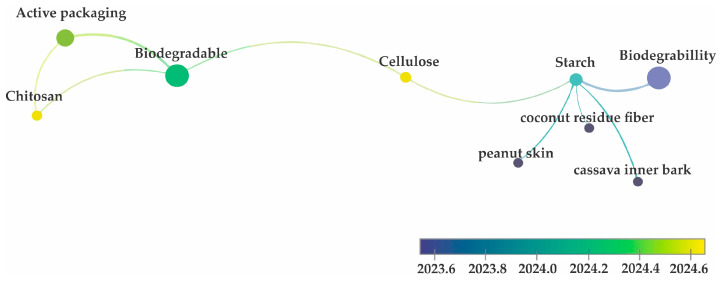
Keyword co-occurrence analysis conducted through the VOSviewer software using the terms ‘biodegradable foams and food packaging’.

**Figure 5 foods-14-04027-f005:**
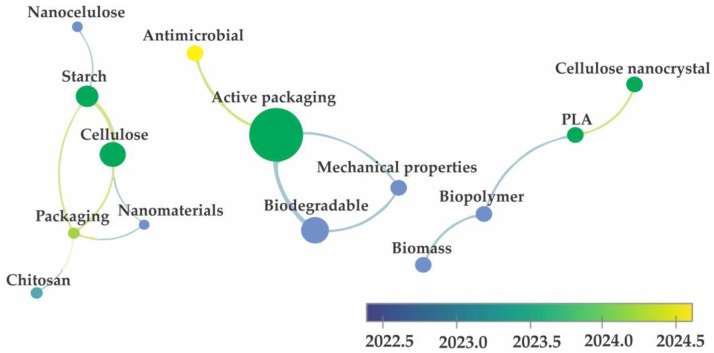
Keyword co-occurrence analysis conducted through the VOSviewer software using the terms ‘starch foams nanocomposites, and food packaging’.

**Figure 6 foods-14-04027-f006:**
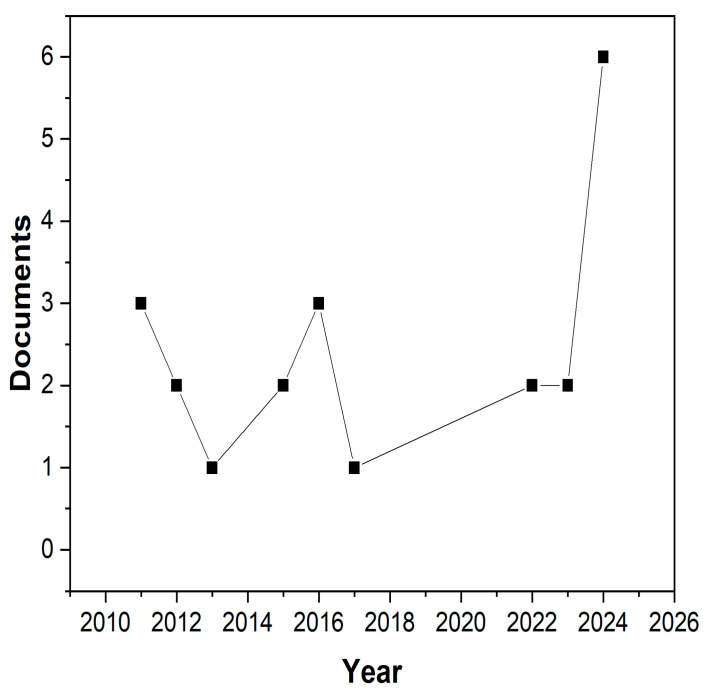
Graph showing the number of articles retrieved from Scopus using the keywords “nanocomposites, food, packaging, biodegradable, and foams” over time.

**Table 1 foods-14-04027-t001:** Comparative table of the advantages and disadvantages of the most commonly used production methods for manufacturing biodegradable foams.

Manufacturing Methods					
Criteria	Thermoforming	3D Printing	Supercritical Fluid	Extrusion	References
Production Cost	Good	Low	Medium	Medium	[33,35]
Large-scale Production	Good	Low	Good	Medium	[33,36,37]
Eco-friendly Production	Good	Medium	Good	Good	[34,38]
Use of Renewable Resources	Medium	Medium	Good	Medium	[18,21]
Cost of Equipment	Medium	Medium	High	Medium	[35]
Complex Shape Capability	High	High	High	Medium	[21,33]
Scalability	High	Low	Low	High	[23,35]
Limitations	Low	High	Medium	Medium	[33,37]

**Table 2 foods-14-04027-t002:** Studies and values of mechanical and hydrophilic analyses of foams made from cassava starch incorporated with plant cellulose fibers.

Materials	Reference	Properties	
Starch with 6% of modified coconut residue fiber + 8% of coconut residue fiber	[60]	Tensile strength (MPa)	Properties related to water interaction
		0.23 ± 0.02	No analysis presented
Starch with 0% of sesame cake and Starch with 40% of sesame cake	[62]	Tensile strength (MPa)	Water absorption capacity
		1.16 ± 0.33 and 0.43 ± 0.04	154% and 141%
Starch with 0% of peanut skin and Starch with 24% of peanut skin	[61]	Tensile strength (MPa)	Hydrophilic character
		1.1 ± 0.1 and 1.0 ± 0.3	63 ± 2° and 94 ± 1°
Starch with 0% of coconut residue fiber and starch with 8% of coconut residue fiber	[59]	Tensile strength (MPa)	Water absorption
		0.14 and 0.16	No values presented
Starch with 0% of cassava inner bark and 50% of starch and 50% of cassava inner bark	[57]	Tensile strength (MPa)	Water absorption
		3.3 ± 0.9 and 1.8 ± 0.6	26% and 14%

**Table 3 foods-14-04027-t003:** Studies and values of mechanical and hydrophilic analyses of foams obtained from composite and nanocomposite materials.

Materials	Reference	Properties	
PLA with 3% of organoclay and carvacrol cocrystal	[83]	Tensile strength (MPa)	Control release and antioxidant activity
		3.8 ± 0.2	
Starch with 0% of bacterial nanocellulose fibers and Starch with 2% of bacterial nanocellulose fibers	[84]	Tensile strength (MPa)	Water absorption capacity
		No analysis presented	No values are presented, but the nanocellulose decreases susceptibility.
Cellulose with 0% of montmorillonite and Cellulose with 10% of montmorillonite	[85]	Tensile strength (MPa)	Water vapor permeability (g·mm/m^2^·h·Pa)
		3.5 ± 0.6 and 7.6 ± 0.4	0.037 ± 0.002 and 0.03 ± 0.001
Rubber, starch and activated carbon	[82]	Tensile strength (MPa)	Ethylene absorbing properties
		0.80 ± 0.11	

## Data Availability

The data that support the findings of this study are available from the corresponding author upon reasonable request. Requests for data access should be directed to Fabrício C. Tanaka (tanaka.fabricio@usp.br).

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
