# Peer review of "The Future of Sustainable Packaging: Exploring Biodegradable Solutions Through Extrusion, Thermo-Expansion, 3D Printing and Supercritical Fluid from Agro-Industry Waste"

_foods, 2025, doi:10.3390/foods14234027_

Round 1

Reviewer 1 Report

Comments and Suggestions for Authors

Brief summary

This review paper gives insight in polymers and methods of preparation of biodegradable packaging foams, as the substitution for expanded polystyrene. Specifically, it reviews the employment of extrusion, thermo-expansion, 3D printing, and supercritical fluid for preparation solid foams from agro-industry waste.

General Comments

Manuscript is clearly written and well organized, with clear sections. Subject is up to date and relevant to the field. However, there is too much attention given on the statistics of the number of published papers and their key words and connections throughout the manuscript. This should be reduced, and these sections should be extended with more examples of investigation or application of particular polymer.

Specific Comments

Line 38: Use words like “two times” or “double” instead of “2X”.

Line 50-59: This paragraph seems excessive and off-topic, since the main subject of the paper is not biology or food waste in particular. The point of plastic waste contamination is already made in previous parts.

Line 162: Table 1 – the last row of table “Limitations” needs to be clarified. What does “good” means in terms of limitations? Also limitations should be more discussed in the text, for example what limitations are specific to each method, because the most of them are already listed thru other criteria in the Table 1, so it is unclear what other limitation authors suggest.

Line 174: Figure 2 – there is “S” missing in polystyrene within the quote next to year 1951.

Line 411: “As presented at Figure 2, along with starch, cellulose is one of the most explored polymers for the production of biodegradable foams” – Figure 2 shows a timeline of milestones, and not the prevalence of biodegradable polymers.

Author Response

Reviewer #1: Line 38: Use words like “two times” or “double” instead of “2X”.

Answer: We would like to thank you for the corrections and suggestions. Regarding the suggested change, the modification has been implemented as two times on line 38, page 1.

Reviewer #1: Line 50-59: This paragraph seems excessive and off-topic, since the main subject of the paper is not biology or food waste in particular. The point of plastic waste contamination is already made in previous parts.

Answer: We agree with the reviewer’s suggestion, and the corresponding passage has been removed from the manuscript.

Reviewer #1: Line 162: Table 1 – the last row of table “Limitations” needs to be clarified. What does “good” means in terms of limitations? Also limitations should be more discussed in the text, for example what limitations are specific to each method, because the most of them are already listed thru other criteria in the Table 1, so it is unclear what other limitation authors suggest.

Answer: There was an input error in this table, and the correct term should be low instead of good. A new section explaining the criterion used in this table has been added in lines 156–162, page 4.

Reviewer #1: Line 174: Figure 2 – there is “S” missing in polystyrene within the quote next to year 1951.

Answer: The correction has been applied to Figure 2.

Reviewer #1: Line 411: “As presented at Figure 2, along with starch, cellulose is one of the most explored polymers for the production of biodegradable foams” – Figure 2 shows a timeline of milestones, and not the prevalence of biodegradable polymers.

Answer: We apologize for the oversight. In this section, the authors were referring to Figure 4. The typographical error has been corrected, and the updated version is now presented on line 410, page 11.

Reviewer 2 Report

Comments and Suggestions for Authors

The manuscript entitled “The Future of Sustainable Packaging: Exploring Biodegradable Solutions Through Extrusion, Thermo-Expansion, 3D Printing and Supercritical Fluid from Agro-Industry Waste” aims to discuss the current and important topic of developing biodegradable polysaccharide-based foams as viable alternatives to expanded polystyrene (EPS). The topic is relevant, well-structured, and clearly organized into chapters. It fits well within the journal's scope. However, the manuscript, in its current form, does not demonstrate sufficient analytical depth or originality to be published without substantial revision. My specific comments are given below.

At the end of the introduction, the authors should explicitly state the paper's novelty. What does this review bring that is new compared to the previous ones? It is only stated that the paper fills a gap, but it is not explained what this gap consists of.

The pictures are not informative enough. Figure 2 is small and unreadable. The other pictures do not contribute to understanding either, they seem more illustrative than informative. I recommend that the authors increase the resolution and size of the figures, add additional, more meaningful graphs, and show a scheme that connects the types of agro-waste and the corresponding polysaccharides.

The tables are uneven and need improvement.

In the section "Future prospects and challenges," the authors mention price and infrastructure, but the discussion remains superficial. It should include an analysis of why innovative solutions in the field of non-plastic packaging are still not widely accepted, who it suits, and who slows down progress. This dimension is crucial for rounding off the topic and for the work to gain critical value.

The manuscript is extensive and often repeats the same information (e.g., the description of extrusion across two separate chapters). The authors should avoid duplication and shorten the narrative, especially in technical descriptions. Some sentences are too descriptive and can be summarized in more concrete conclusions.

Abbreviations should only be defined on first occurrence and then used consistently throughout the manuscript.

In the conclusion, practical guidelines for future research should be more clearly highlighted.

Author Response

Reviewer #2: At the end of the introduction, the authors should explicitly state the paper's novelty. What does this review bring that is new compared to the previous ones? It is only stated that the paper fills a gap, but it is not explained what this gap consists of.

Answer: This review brings practical guidelines for material selection, processing routes, and research priorities aimed at accelerating the development and adoption of biodegradable foamed packaging. This combined approach has not been addressed in previous reviews and directly clarifies the specific gap that this work fills in the current literature. To better emphasize the objective of this work, the final paragraph of the introduction has been revised accordingly and can now be found in lines 109–125, page 3.

Reviewer #2: The pictures are not informative enough. Figure 2 is small and unreadable. The other pictures do not contribute to understanding either, they seem more illustrative than informative. I recommend that the authors increase the resolution and size of the figures, add additional, more meaningful graphs, and show a scheme that connects the types of agro-waste and the corresponding polysaccharides.

Answer: Based on the reviewer’s suggestions, we enhanced the quality of all figures and strengthened the connection between agro-industrial residues and starch-based foams in Figure 4. We also incorporated additional explanatory text in lines 341-346, p.9; 488-505, p.13; 538-541, p.15, to make the figures more informative and improve their overall clarity.

Reviewer #2: The tables are uneven and need improvement.

Answer: Tables 2 and 3 have been modified in an effort to standardize their dimensions.

Reviewer #2: In the section "Future prospects and challenges," the authors mention price and infrastructure, but the discussion remains superficial. It should include an analysis of why innovative solutions in the field of non-plastic packaging are still not widely accepted, who it suits, and who slows down progress. This dimension is crucial for rounding off the topic and for the work to gain critical value.

Answer: Future prospects and challenges section was included as a concluding element aimed at outlining the main challenges that this research field is expected to face, as well as the barriers associated with the current packaging market. In this section, we propose several potential research directions and incentives to stimulate further advancements. Although our research group has extensive experience with biodegradable packaging and recognizes its importance as a long-term strategy, we also acknowledge the difficulty of competing against a well-established market based on petroleum-derived materials, which still offer clear performance advantages and benefit from large-scale industrial consolidation. We also understand that significant progress in this area will require substantial investment, which may primarily benefit larger companies and more economically developed regions.

However, introducing a more critical or confrontational tone in this section could alter its intended purpose and risk being interpreted by some readers as accusatory. For this reason, and to maintain the constructive nature of the manuscript, this is the only suggestion that we respectfully chose not to implement.

Reviewer #2: The manuscript is extensive and often repeats the same information (e.g., the description of extrusion across two separate chapters). The authors should avoid duplication and shorten the narrative, especially in technical descriptions. Some sentences are too descriptive and can be summarized in more concrete conclusions.

Answer: We thank the reviewer for the suggestion. Based on this comment, we removed the passages referring to the year in which extrusion was first applied to foam production, which previously appeared in Section 2 and Section 2.2. It is important to note that extrusion was originally described in two sections in order to first present the foaming techniques in chronological order in Section 2, and then provide a more detailed discussion in Section 2.2. Additional deletions were made following the same rationale, including the former lines 50–59.

Reviewer #2: Abbreviations should only be defined on first occurrence and then used consistently throughout the manuscript.

Answer: Fixed.

Reviewer #2: In the conclusion, practical guidelines for future research should be more clearly highlighted.

Answer: To address the reviewer’s suggestion, the penultimate paragraph of the conclusion has been revised to place greater emphasis on practical guidance for future research in this field. This updated paragraph is now presented in lines 632–643, page 17.

Reviewer 3 Report

Comments and Suggestions for Authors
  • The manuscript is well-structured, clearly written, and organized. However, the database used in the analysis appears to need updating. For instance, in Figure 6, the graph seems inconsistent—there is only one study shown for 2017. Could the authors please verify and update the data? I found, for example, a 2015 study titled “Nanoporous cellulose nanocomposite foams as high insulated food packaging materials.” I recommend the authors review the literature again and update the figures and related discussions accordingly.
  • Line 64: Please provide a supporting reference for the statement, “Currently, 56 major global brands are responsible for more than 50% of plastic waste.”

  • Please research the literature using the relevant keywords and update the plot and table accordingly. Adding more previous research work in the discussion section would improve the depth and currency of the manuscript.

Author Response

Reviewer #3: The manuscript is well-structured, clearly written, and organized. However, the database used in the analysis appears to need updating. For instance, in Figure 6, the graph seems inconsistent—there is only one study shown for 2017. Could the authors please verify and update the data? I found, for example, a 2015 study titled “Nanoporous cellulose nanocomposite foams as high insulated food packaging materials.” I recommend the authors review the literature again and update the figures and related discussions accordingly.

Answer: First, we would like to thank you for your suggestions. As instructed, we reviewed the keywords, and the 2015 article has been included in the dataset, increasing the number of studies from 7 to 22.

Reviewer #3: Line 64: Please provide a supporting reference for the statement, “Currently, 56 major global brands are responsible for more than 50% of plastic waste.

Answer: In this section, we have added a reference to support this information.

Reviewer #3: Please research the literature using the relevant keywords and update the plot and table accordingly. Adding more previous research work in the discussion section would improve the depth and currency of the manuscript.

Answer: Based on the first recommendation, we updated the number of articles involving nanocomposite foams. This update was also applied to the searches and keywords used to generate Figure 5. These revisions can be found in lines 341–346 (page 9) and lines 538–541 (page 15). Additionally, a new paragraph providing a more in-depth discussion of research on nanocomposite foams has been included in lines 488–505 (page 13).

Round 2

Reviewer 2 Report

Comments and Suggestions for Authors

The figures still appear low-quality. They are not very informative and are not visually appealing. In their current form, they do not attract attention. It would be helpful if the authors could improve the figures to make them clearer and more engaging.